# Effects of Pig Manure and Its Organic Fertilizer Application on Archaea and Methane Emission in Paddy Fields

**Jianqiang Wu [1], Min Wang [1], Peng Li [2], Leyang Shen [3], Mingyi Ma [4], Boyu Xu [1], Shuyuan Zhang [5], Chenyan Sha [1], Chunmei Ye [1], Lijun Xiong [1] and Shenfa Huang [1,2,\*]**

[1]  Shanghai Academy of Environmental Sciences, Shanghai 200233, China; wujq@saes.sh.cn (J.W.); wangm@saes.sh.cn (M.W.); octxu.cn@outlook.com (B.X.); shacy@saes.sh.cn (C.S.); ycm_1995@163.com (C.Y.); xionglj@saes.sh.cn (L.X.)
[2]  College of Environmental Science and Engineering, Donghua University, Shanghai 201600, China; lipeng951204@163.com
[3]  Shanghai Key Laboratory of Materials Protection and Advanced Materials in Electric Power, Shanghai University of Electric Power, Shanghai 200090, China; leyangshen@163.com
[4]  Shanghai Science and Technology Museum Station, Shanghai 200231, China; mamy@sstm.org.cn
[5]  School of Perfume and Aroma Technology, Shanghai Institute of Technology, Shanghai 201418, China; zhangshuyuan1101@163.com
\*  Correspondence: saeshuangsf67@163.com; Tel.: +86-6408-5119-2321

**Abstract:** Paddy fields account for 10% of global $CH_4$ emissions, and the application of manure may increase $CH_4$ emissions. In this study, high-throughput sequencing technology was used to investigate the effects of manure application on $CH_4$ emissions and methanogens in paddy soil. Three treatments were studied: a controlled treatment (CK), pig manure (PM), and organic fertilizer (OF). The results showed that the contents of Zn, Cr and Ni in paddy soil increased with the application of manure, but the contents of heavy metals gradually decreased with the growth of rice. The Shannon index and Ace index showed that the application of pig manure and organic fertilizer less affected the diversity and richness of soil Archaea. The results of community composition analysis showed that *Methanobacterium*, *Methanobrevibacter*, *Methanosphaera*, *Methanosarcina* and *Rice_Cluster_I* were the main methanogens in paddy soil after manure and organic fertilizer application. Soil environmental factors were changed after applied manure, among which total potassium (TK) and total nitrogen (TN) were the main environmental factors affecting methanogens in paddy soil. The changes of soil environmental factors affected the community composition of methanogens, and the increase of the relative abundance of methanogens maybe the main reason for the increase of $CH_4$ emission flux. The relative abundance of methanogens and $CH_4$ emission flux in paddy soil were increased by both pig manure and organic fertilizer application, and pig manure had a bigger impact than organic manure.

**Keywords:** pig manure; organic fertilizer; paddy soil; $CH_4$; methanogens

## 1. Introduction

Methane ($CH_4$), as a primary global greenhouse gas concern, has a significant impact on the carbon and nitrogen cycles in the global ecosystem, and on climate change. Studies show that the greenhouse effect of methane is 21 times higher than that of $CO_2$ in the next 100 years, accounting for 25% of the total contribution of greenhouse gases in 2021 [1,2]. The increase of man-made greenhouse gas emissions intensifies the global warming, and it is considered the second-largest source of greenhouse gas emissions after cities. As the most important food crop in the world, the growth process of rice is closely related to methane emission, which plays a key role in the process of global climate change. China's rice field area accounts for about 23% of the total rice planted globally, ranking second in the world. Therefore, as one of the primary sources of greenhouse gas emissions, paddy fields in China have drawn widespread attention from scholars at home and abroad.

There is a rapid production of livestock and poultry manure in China every year as these industries develop, with the prediction that induced $CH_4$ emissions will increase 60% by 2030. The amount of $CH_4$ emissions produced by paddy soil is varied in different planting systems, crop types, and with different fertilization use and water management, among which fertilizers are a key factor [3]. The application of pig manure and organic fertilizer may bring heavy metals into the paddy soil. Meanwhile, it may affect the archaeal community and ultimately influence the emission of $CH_4$. Studies have shown that the application of organic fertilizers can exacerbate $CH_4$ emissions in paddy fields. However, a well-proportioned number of fertilizers can ameliorate the soil carbon pool or reduce $CH_4$ emissions and its warming potential [4]. The study of the effect of different fertilization measures on the influence of $CH_4$ emission characteristics and its microbiological mechanism in double cropping rice field, which was written by Tang et al. [3], found that under the different fertilization treatments, the $CH_4$ emission flux was noticeably higher than those of the control group in both early and late growth stages, and the combined use of organic and inorganic fertilizers facilitates the emergence functional microorganisms in the paddy ecosystem. Organic matter in farmland can contribute to global warming; Das et al. [5] revealed the incorporation of rice straw and livestock manure in farmland can trigger the release of greenhouse gas to the atmosphere, despite their positive impact on crop yield and soil fertility. The similar research results from Ma and Wang [6,7] showed that the usage of organic fertilizer could potentially increase $CH_4$ emissions in paddy fields. Among all $CH_4$ emission sources, about 69% of $CH_4$ comes from the metabolic activities of methanogens. Therefore, methanogens are the most important biogenic source of methane production. At present, a great number of studies have focused on the effects of different fertilization measures on $CH_4$ emission, methane oxidation activity and archaeal community composition in paddy fields [4–8]; yet the studies fail to take into account the microbial mechanisms of changes in soil methanogen community structure on $CH_4$ emission flux after manure and organic fertilizer application. The application of manure to soil can noticeably increase the number and activity of Archaea and provide many nutrients for $CH_4$ production; also, the composition and diversity of Archaea is altered, along with variation in the soil environment, after manure input. As a result, this changes the activities of methanogens and metanotrophs, and consequently affects the $CH_4$ emission flux in soil.

Therefore, to analyze the chronic effect of manure application on $CH_4$ and soil microorganisms, this research is based on an experimental paddy field in Chongming Island, Shanghai, in which an analysis of $CH_4$ emissions and key functional microorganisms is introduced, to provide theoretical basis and data support for resource utilization of livestock manure and the effects of manure fertilizer on $CH_4$ emission flux in paddy fields and key functional microorganisms.

## 2. Materials and Methods

### 2.1. Field Experimental Condition

In this paper, field tests and observations of various indexes were carried out on the experimental field in 0.03 $km^2$, located within the Experimental Base of Farmland Soil Ecological Process in Zhongxing Town, Chongming Island (121°09′30″~121°54′00″ E, 31°27′00″~31°5l′15″ N), Shanghai, China, which was established by Shanghai Academy of Environmental Sciences. The local climate is the typical subtropical oceanic monsoon climate of north Asia, with 229 frost-free days of the year. This location is the main crop-growing area of Shanghai. The experimental base was built on the local farmland, with the same farming conditions and environment as the surrounding area.

### 2.2. Design of the Experiment

(1) Selection and source of livestock and poultry manure

Pig manure, the typical livestock and poultry manure, and the organic manure made from Shanghai were selected as the combinations of manure fertilizer, which were all taken from a pig farm on Chongming Island. The annual output of the pig farm was about

26,000 pigs, and the annual output of manure was about 20,000 tons. Swine manure is collected and dried, while organic fertilizer is made from the same species of pig manure through conventional anaerobic thermophilic composting.

(2) Experiment bases setting

Three treatment groups were set up in the field experiment (Figure 1), which were a controlled group (CK), pig manure (PM), and organic fertilizer (OF). Each group was set up with three parallel plots, and each plot covered an area of 9 m² (3 m × 3 m). To avoid mutual interference between samples, 0.5 mm anti-seepage film was used to block each sample, and the buried depth of anti-seepage film was 40 cm. A 20 cm footpath was set up between each sample site for on-site monitoring and sample collection. According to the rice growth cycle, a total of three sampling times were set, namely heading stage, flowering stage, and mature stage.

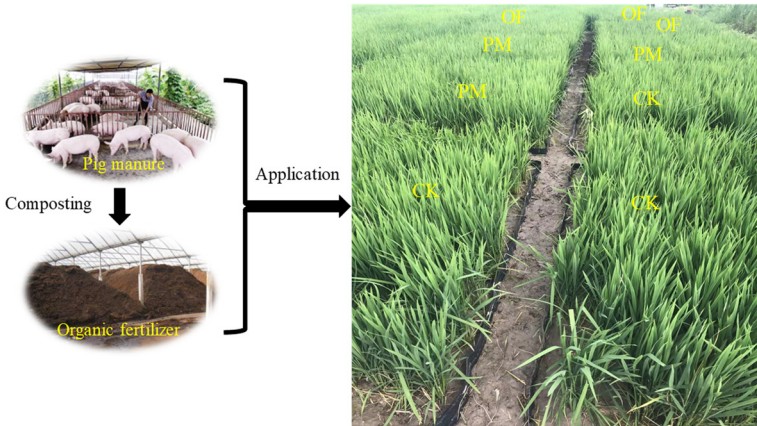

**Figure 1.** Site layout (CK-controlled group, PM-pig manure, OF-organic fertilizer).

(3) Crop selection and fertilization

The crop used for the experiment was rice, a common local crop, planted by transplanting seedlings. The dosage of manure was 2 kg·m⁻² in accordance with the Technology code for land application rates of livestock and poultry manure (GB-T 25246-2010). The manure will be applied to the soil as a base fertilizer, and then plowed after fertilization. Seedlings were transplanted seven days after fertilization, and the transplanting time was in early June 2020. Other management measures were the same as those in general paddy fields.

### 2.3. Methods for Collecting and Analyzing Soil Samples

2.3.1. Collection of Soil Sample

(1) Sampling time: According to the growth cycle of rice, three times of sampling are set, which were in late August (heading stage—CK3, PM3, OF3), early September (flowering stage—CK4, PM4, OF4) and mid-October (mature stage—CK5, PM5, OF5).

(2) Sampling method: A five-point sampling method was used to collect 500 g soil per plot at 0–20 cm deep, repeated three times.

(3) Sample storage and transportation: After removing debris, earthworms and plant residues, the sample was divided into four parts. One part of the sample was placed in a sterile Eppendorf (EP) tube, the other part in a sterile brown bottle. The rest was placed in two self-sealed bags and transported to each laboratory in a refrigerated box for microbiological and physiochemical properties detection.

2.3.2. Analysis of Soil Physical and Chemical Properties

After natural weathering, the soil samples were ground, sifted by 10-mesh and 100-mesh, and stored below 4 °C; three identical treatments for each group of samples.

Soil pH measurement: A LEICI PHSJ-5 acidometer (Shanghai Kexiao Scientific Instrument Co., Ltd., Shanghai, China) was used to measure the pH. Soil samples of 4 g were

weighed, and 10 mL sterile water was added, according to the soil-water ratio of 1:2.5 (mass concentration). After it was put into a shaker and shaken for 0.5 h (speed: 200 r·min$^{-1}$; temperature: 28 °C), the clear liquid after filtration was measured on the machine.

Soil nutrient determination: Soil organic matter was determined by potassium dichromate titration. Total nitrogen (TN) was determined by a CN 802 carbon and nitrogen element analyzer (Beijing Legoltech Technology Co., Ltd., Beijing, China) (dry firing method). Total phosphorus (TP) was determined by a UV spectrophotometer (T6- New Century, Shanghai Yuantong Instrument Co., Ltd., Shanghai, China). Total potassium (TK) was determined by a WGH6400 flame photometer (Shanghai Changxi Instrument Co., Ltd., Shanghai, China). Soil ammonia nitrogen (NN) was determined by the HJ 634-2012 method (Beijing Legoltech Technology Co., Ltd., Beijing, China). Soil nitrate nitrogen (AN) was determined by NY/T 1116-2014 (Beijing Legoltech Technology Co., Ltd., Beijing, China). The temperature(T) and conductivity (MC) were measured by a portable soil three parameter tester (Beijing Rike Huize Technology Co., Ltd., Beijing, China).

Determination of heavy metals: Determination of As and Hg was performed using an atomic fluorescence spectrophotometer (AFS-3100, Beijing Baode Instrument Co., Ltd., Beijing, China); determination of Pb and Cd was done by using an atomic absorption spectrophotometer (TAS-990, Shanghai Baihe Instrument Technology Co., Ltd., Shanghai, China); flame atomic absorption spectrophotometry (HJ 491-2019, Shanghai Baihe Instrument Technology Co., Ltd., Shanghai, China) was used to determine Cu, Zn, Ni and Cr.

### 2.3.3. Analysis of Soil Microorganism

(1) DNA extraction and PCR amplification

First, microbial DNA was extracted from soil samples by a Fast DNA Spin Kit (MP Biomedicals), and then the final DNA concentration and purity were determined using a NanoDrop 2000 UV-visible spectrophotometer (Thermo Scientific, Wilmington, Waltham, MA, USA). The hypervariable region V3–V4 of the Archaea 16S rRNA gene was amplified with primer pairs 524F10extF_Arch958RmodR 524F10extF (TGYCAGCCGCCGCGGTAA) and Arch958RmodR (YCCGGCGTTGAVTCCAATT 434) by an ABI GeneAmp$^®$ 9700 PCR thermocycler (ABI, CA, California, USA). The PCR amplification of the 16S rRNA gene was performed as follows: initial denaturation at 95 °C for 3 min, followed by 27 cycles of denaturing at 95 °C for 30 s, annealing at 55 °C for 30 s, and extension at 72 °C for 45 s, and single extension at 72 °C for 10 min, and end at 4 °C. The PCR mixtures contain 5 × TransStart FastPfu buffer 4 μL, 2.5 mM dNTPs 2 μL, forward primer (5 μM) 0.8 μL, reverse primer (5 μM) 0.8 μL, TransStart FastPfu DNA Polymerase 0.4 μL, template DNA 10 ng, and finally ddH$_2$O up to 20 μL. PCR reactions were performed in triplicate. The PCR product was extracted from 2% agarose gel and purified using the AxyPrep DNA Gel Extraction Kit (Axygen Biosciences, Union City, CA, USA), according to manufacturer's instructions, and quantified using a Quantus™ Fluorometer (Promega, Madison, WI, USA).

(2) Illumina MiSeq sequencing

Purified amplicons were pooled in equimolar, and paired-end sequenced on an Illumina MiSeq PE300 platform/NovaSeq PE250 platform (Illumina, San Diego, CA, USA) according to the standard protocols by Majorbio Bio-Pharm Technology Co. Ltd. (Shanghai, China). The raw reads were deposited into the NCBI Sequence Read Archive (SRA) database.

(3) Processing of sequencing data

The raw 16S rRNA gene sequencing reads were demultiplexed, quality-filtered by fastp version 0.20.0 [9], and merged by FLASH version 1.2.7 [10] with the following criteria: (i) the 300 bp reads were truncated at any site receiving an average quality score of <20 over a 50 bp sliding window, and the truncated reads shorter than 50 bp were discarded, and reads containing ambiguous characters were also discarded; (ii) only overlapping sequences longer than 10 bp were assembled according to their overlapped sequence. The maximum mismatch ratio of overlap region is 0.2. Reads that could not be assembled were

discarded; (iii) samples were distinguished according to the barcode and primers, and the sequence direction was adjusted.

Operational taxonomic units (OTUs) with 97% similarity cutoff [11,12] were clustered using UPARSE version 7.1 [11], and chimeric sequences were identified and removed. The taxonomy of each OTU representative sequence was analyzed by RDP Classifier version 2.2 [13] against the 16S rRNA database (Silva v138/16s_archaea), using a confidence threshold of 0.7.

### 2.3.4. Determination of $CH_4$ Mission Flux

A Polyvinyl chloride (PVC) soil breathing collar was embedded in each parallel plot, with a diameter of 20 cm and a height of 12 cm. After rice was transplanted, it was embedded into the soil at a depth of 10 cm, and the upper end was 2 cm above the ground. To avoid experimental error, the base is embedded in the gap between the plant, avoiding their roots. To eliminate the influence of plant photosynthesis and ground litter on the data, the plots were cleaned the day before each measurement. The portable trace $N_2O/CH_4$ soil flux measurement system (N$_2$OM1-919) was used for the determination of $CH_4$, and the measurement time was also selected between 10:00 and 17:00 on sunny days. The measurement frequency was 1HZ, and one measurement cycle was 5 min. Two measurements were made at one time, and was repeated three times. Each measurement was conducted on the same day.

### 2.4. Statistical Analysis

Microsoft Excel 2016 was used to calculate the mean and standard deviation of the original data. Circos was drawn using circos-0.67-7 (http://circos.ca/, accessed on 9 December 2020), and principal component analysis used R (Version 3.6.3) for PCA statistical analysis and plotting. One-way ANOVA (SPSS V. 24) was used to analyze the significance of differences in soil physiochemical factors, Archaea diversity index, and the difference between groups. SPSS 24 Spearman was used to analyze the correlation between environmental factors and microorganisms.

## 3. Results

### 3.1. Soil Physiochemical Properties in Paddy Field Applied with Manure

The changes of heavy metal content in paddy soil after manure application are shown in Table 1. At the heading stage, the content of Zn in the OF group was 9.39% higher than that in the CK group ($p < 0.05$), reaching 77.67 mg·kg$^{-1}$; at the flowering stage, the heavy metal content of different treatment groups had little difference. At the mature stage, the contents of As and Cr in the CK group were higher than those in the fertilization group, and significantly higher than those in the PM group ($p < 0.05$), 21.62% and 41.17% higher, respectively, up to 5.40 mg·kg$^{-1}$ and 104.00 mg·kg$^{-1}$. With the growth of rice, the content of heavy metals in the soil constantly decreases, which may be due to the conversion of heavy metal ions in the soil into a stable state through adsorption, ion exchange and complex precipitation after the application of manure, and then reduce the bioavailability and migration of heavy metals in the soil [14]. At the heading stage, the contents of Cr and Ni in the PM group were the highest, at 188.00 mg·kg$^{-1}$ and 112.00 mg·kg$^{-1}$, respectively. The high content of heavy metals in the fertilization group may be attributed to the heavy metals in the manure brought into the soil after the application of manure.

**Table 1.** Effects of manure application on heavy metals in paddy soil (mg·kg$^{-1}$).

| Period | Simple | Cu | Pb | As | Cr | Hg | Zn | Cd | Ni |
|---|---|---|---|---|---|---|---|---|---|
| | CK3 | 20.33 ± 0.58 a | 20.20 ± 1.97 a | 6.37 ± 1.40 a | 110.67 ± 59.28 a | 0.060 ± 0.010 a | 71.00 ± 2.65 b | 0.103 ± 0.006 a | 70.33 ± 18.58 a |
| Heading stage | PM3 | 20.00 ± 2.00 a | 20.03 ± 1.01 a | 5.83 ± 1.08 a | 188.00 ± 44.53 a | 0.054 ± 0.007 a | 74.00 ± 1.73 ab | 0.107 ± 0.012 a | 112.00 ± 27.62 a |
| | OF3 | 22.33 ± 2.89 a | 18.67 ± 0.65 a | 5.19 ± 0.78 a | 158.67 ± 73.28 a | 0.051 ± 0.007 a | 77.67 ± 3.79 a | 0.100 ± 0.000 a | 92.00 ± 31.24 a |
| | CK4 | 18.00 ± 1.00 a | 15.07 ± 2.05 a | 5.36 ± 0.26 a | 65.67 ± 13.20 a | 0.050 ± 0.018 a | 72.67 ± 3.21 a | 0.110 ± 0.010 a | 51.00 ± 23.39 a |
| Flowering stage | PM4 | 18.00 ± 1.00 a | 16.73 ± 3.42 a | 4.91 ± 0.06 a | 71.33 ± 17.39 a | 0.038 ± 0.061 a | 63.33 ± 21.22 a | 0.110 ± 0.000 a | 42.67 ± 9.07 a |
| | OF4 | 17.33 ± 1.15 a | 20.37 ± 3.92 a | 4.26 ± 1.12 a | 87.67 ± 8.33 a | 0.037 ± 0.002 a | 67.00 ± 2.65 a | 0.100 ± 0.060 a | 49.67 ± 4.93 a |
| | CK5 | 20.33 ± 2.52 a | 15.50 ± 1.77 a | 5.40 ± 0.55 a | 104.00 ± 21.93 a | 0.065 ± 0.020 a | 59.00 ± 6.56 a | 0.107 ± 0.012 a | 72.00 ± 25.53 a |
| Mature stage | PM5 | 19.67 ± 0.58 a | 14.57 ± 0.78 a | 4.44 ± 0.55 b | 73.67 ± 8.50 b | 0.069 ± 0.020 a | 61.67 ± 3.79 a | 0.110 ± 0.000 a | 50.67 ± 2.31 a |
| | OF5 | 18.33 ± 0.58 a | 14.03 ± 0.90 a | 4.61 ± 0.11 ab | 83.00 ± 7.55 ab | 0.047 ± 0.015 a | 58.67 ± 6.66 a | 0.107 ± 0.006 a | 54.67 ± 10.69 a |

Different lowercase letters indicate significant differences between groups *(p < 0.05)*.

Manure increased the soil organic matter content (Table 2), and at the heading stage, the organic matter content of the PM group was significantly higher than that of the OF and CK groups ($p < 0.05$), both being 5.00 g·kg$^{-1}$ higher, reaching 16.49%; at the flowering stage, the soil organic matter content of both PM and OF groups was higher than that of the CK group; both were higher by 3.00 g·kg$^{-1}$, reaching 18.37%. Overall, the application of pig manure and organic fertilizer had little effect on soil TN content at the heading and flowering stages of rice, neither of which reached significant levels; however, at the mature stage, the OF group had significantly higher TN content than the CK and PM groups ($p < 0.05$) by 43.62% and 56.98% to 1.35 g·kg$^{-1}$, respectively. At the heading stage, the soil TP content was ranked as OF = PM > CK, and the OF and PM groups were significantly higher than that of the CK group ($p < 0.05$), both being 18.00% higher and reaching 1.18 g·kg$^{-1}$. At the flowering and mature stage, the soil TP content was ranked as OF > PM > CK, among which, the OF group was significantly higher than that of the CK group at the flowering stage ($p < 0.05$), being 17.50% higher and reaching 1.41 g·kg$^{-1}$. At the heading stage, the PM group had significantly higher TK content than the OF group ($p < 0.05$), which was 3.10 g·kg$^{-1}$ higher. At the flowering stage, the OF group had higher TK content, which was noticeably higher than the CK and PM groups, reaching 21.67 g·kg$^{-1}$, respectively; at the mature stage of rice, there was no significant difference in soil TK content. Overall, the application of pig manure and organic fertilizer during the rice growth cycle transiently increased the soil TK content.

**Table 2.** Effects of fertilization on soil pH and nutrients in paddy soil.

| Period | Simple | pH (-) | Organic Matter (OM) (g·kg$^{-1}$) | Total Nitrogen (TN) (g·kg$^{-1}$) | Total Phosphorus (TP) (g·kg$^{-1}$) | Total Potassium (TK) (g·kg$^{-1}$) |
|---|---|---|---|---|---|---|
| | CK3 | 8.4 ± 0.06 a | 30.33 ± 3.06 b | 1.35 ± 0.06 a | 1.00 ± 0.13 b | 25.90 ± 1.13 ab |
| Heading stage | PM3 | 8.32 ± 0.03 a | 35.33 ± 0.58 a | 1.46 ± 0.09 a | 1.18 ± 0.03 a | 27.80 ± 1.23 a |
| | OF3 | 8.23 ± 0.04 b | 30.33 ± 0.58 b | 1.46 ± 0.07 a | 1.18 ± 0.05 a | 24.70 ± 1.42 b |
| | CK4 | 8.31 ± 0.28 a | 16.33 ± 0.58 a | 0.81 ± 0.24 a | 1.20 ± 0.14 b | 14.77 ± 3.74 b |
| Flowering stage | PM4 | 8.17 ± 0.10 a | 19.33 ± 3.06 a | 0.97 ± 0.13 a | 1.31 ± 0.08 ab | 13.27 ± 2.11 b |
| | OF4 | 8.30 ± 0.18 a | 19.33 ± 2.52 a | 1.10 ± 0.13 a | 1.41 ± 0.08 a | 21.67 ± 3.76 a |
| | CK5 | 8.23 ± 0.02 a | 23.33 ± 2.08 a | 0.94 ± 0.06 b | 1.01 ± 0.07 b | 22.00 ± 3.03 a |
| Mature stage | PM5 | 8.22 ± 0.03 a | 20.33 ± 3.51 a | 0.86 ± 0.05 b | 1.11 ± 0.04 b | 22.27 ± 3.88 a |
| | OF5 | 8.13 ± 0.01 b | 20.33 ± 3.06 a | 1.35 ± 0.26 a | 1.23 ± 0.06 a | 23.23 ± 0.32 a |

Different lowercase letters indicate significant differences between groups ($p < 0.05$).

Soil nutrient content is the material basis of rice growth, and its content varies with the type of fertilization. There are many investigations on the effect of fertilization on soil nutrient content in farmland. For example, the study of Cheng et al. [15] found that long-term application of organic fertilizer not only increased soil organic matter content, but also promoted the conversion of soil available phosphorus. The study of Xia et al. [16] showed that long-term unbalanced fertilization could lead to serious nutrient deficiency and then affect crop growth, while combined application of organic and inorganic fertilizers could accumulate soil nutrients. The contents of N, P and K in soil were also different after different fertilization treatments. The study of Li et al. [17] showed that TN was an important index to measure the nitrogen supply of the soil and was associated with the accumulation and decomposition of soil organic matter. This research indicated that manure application would temporally increase the soil organic matter content of rice fields, which reached the highest at the heading stage, and gradually decreased with the growth of rice, which might be due to the large amount of nutrients consumed by the growth and development of rice. The increase of TP content in soil after application of pig manure and organic fertilizer may be associated with the high content of soluble phosphorus in pig manure.

### 3.2. Bacterial Richness and Diversity of Soil Archaea in Paddy Field Applied with Manure

In this paper, the Shannon diversity index (Shannon), the colony abundance index (Ace), and the sequencing depth index (Coverage) reflect the diversity of soil microbial communities, community richness, and coverage of the sample library in rice fields, respectively (Table 3). The Coverage indices of different treatment groups ranged from 0.9964 to 0.9972, indicating that the sequencing depth of soil samples was reasonable. The effects of different manure applications on the diversity and abundance of archaeal microorganisms in paddy fields were small and did not differ significantly at different periods of rice growth. At the heading stage, the Shannon index was lower in the PM and OF groups than in the CK group, lower than 0.1822 and 0.0727, respectively. By the flowering stage, both groups were higher than the CK group; it was 0.3166 higher in PM group and 0.2144 higher in OF group. Finally, at the mature stage, the PM group was the highest, at 3.2141, the CK group was the second highest, at 3.1943, and the PM group was the lowest, at 3.1452. The soil Ace index of the paddy field was reduced after manure application. At the heading stage, the CK group was 17.6986 and 74.9982 higher than the PM group and the OF group, respectively; only at the flowering stage was the Ace index of the PM group higher than that of the CK group, being 3.3755 higher. Finally, at the mature stage, the CK group was 2.7794 and 7.7883 higher than the PM group and the OF group, respectively. The results indicated that the diversity and abundance of Archaea in the paddy field were less affected after manure application.

**Table 3.** Indexes of archaeal community diversity in paddy fields under different fertilization treatments.

| Period | Simple | Shannon | Ace | Coverage |
|--------|--------|---------|-----|----------|
| Heading stage | CK3 | 3.5197 ± 0.1467 a | 428.4023 ± 75.3949 a | 0.9966 ± 0.0004 a |
| | PM3 | 3.3375 ± 0.2366 a | 410.7037 ± 65.4228 a | 0.9968 ± 0.0005 a |
| | OF3 | 3.4470 ± 0.2200 a | 353.4041 ± 58.3528 a | 0.9972 ± 0.0005 a |
| Flowering stage | CK4 | 3.3291 ± 0.2312 a | 443.5142 ± 89.6234 a | 0.9964 ± 0.0007 a |
| | PM4 | 3.5109 ± 0.1334 a | 446.8897 ± 62.7242 a | 0.9964 ± 0.0004 a |
| | OF4 | 3.4087 ± 0.1180 a | 371.8767 ± 98.2980 a | 0.9968 ± 0.0009 a |
| Mature stage | CK5 | 3.1943 ± 0.2263 a | 361.6134 ± 64.1589 a | 0.9971 ± 0.0005 a |
| | PM5 | 3.2141 ± 0.0532 a | 358.8340 ± 43.6350 a | 0.9970 ± 0.0001 a |
| | OF5 | 3.1452 ± 0.1398 a | 353.8251 ± 104.0263 a | 0.9970 ± 0.0007 a |

Different lowercase letters indicate significant differences between groups ($p < 0.05$).

### 3.3. Taxonomic Composition of Soil Archaeal Microbial Communities in Paddy Field Applied with Manure

The dominant phylum and genus of soil Archaea from paddy fields after application of pig manure and its organic fertilizer are shown in Figure 2; the phylum with relatively high abundance at the phylum level are *Crenarchaeota*, *Halobacterota*, *Thermoplasmatota*, *Altiarchaeota*, *Euryarchaeota*, and *unclassified_k__norank_d__Archaea*, consistent with the results of Wang et al. [18].

The phylum with the highest relative abundance in different treatment groups was *Crenarchaeota*, in agreement with the results of Chen's study [19], but its relative abundance did not differ significantly at different periods. The relative abundance of *Halobacterota* was highest in the OF group at the heading stage with 23.0%, which was 7.0% and 16.8% higher than the CK and PM groups; it was highest in the PM group at the flowering stage, with 13%, which was 2.0% and 8.5% higher than the CK and OF groups. Eventually, at the mature stage, the PM group was still the highest, but decreased by 1.0% compared to the flowering stage, while the difference between the CK and OF groups was smaller. The relative abundance of *Thermoplasmatota* in the CK group decreased with the growth of the rice, being 15.0% at heading stage, while being 10.0% and 11.0% at flowering and mature stage, respectively. The PM group decreased with the growth of rice and was 9.7% at the heading stage, being 1.1% and 3.7% higher than that at the flowering and mature

stage; conversely, the OF group showed a trend of increasing and then decreasing, with the highest being 19.0% at the flowering stage, 3.0% higher than that at the heading stage and 14.3% higher than that at the mature stage. The relative abundance of *Euryarchaeota* in the OF group was highest at 15.0% at the heading stage, 5.5% and 5.3% higher than that in the CK and PM groups, and increased to 16.0% at the flowering stage, 4.0% and 10.0% higher than that in the CK and OF groups; finally, at the mature stage, the PM group was 9.9% and 5.0% higher than that in the CK and OF groups, respectively.

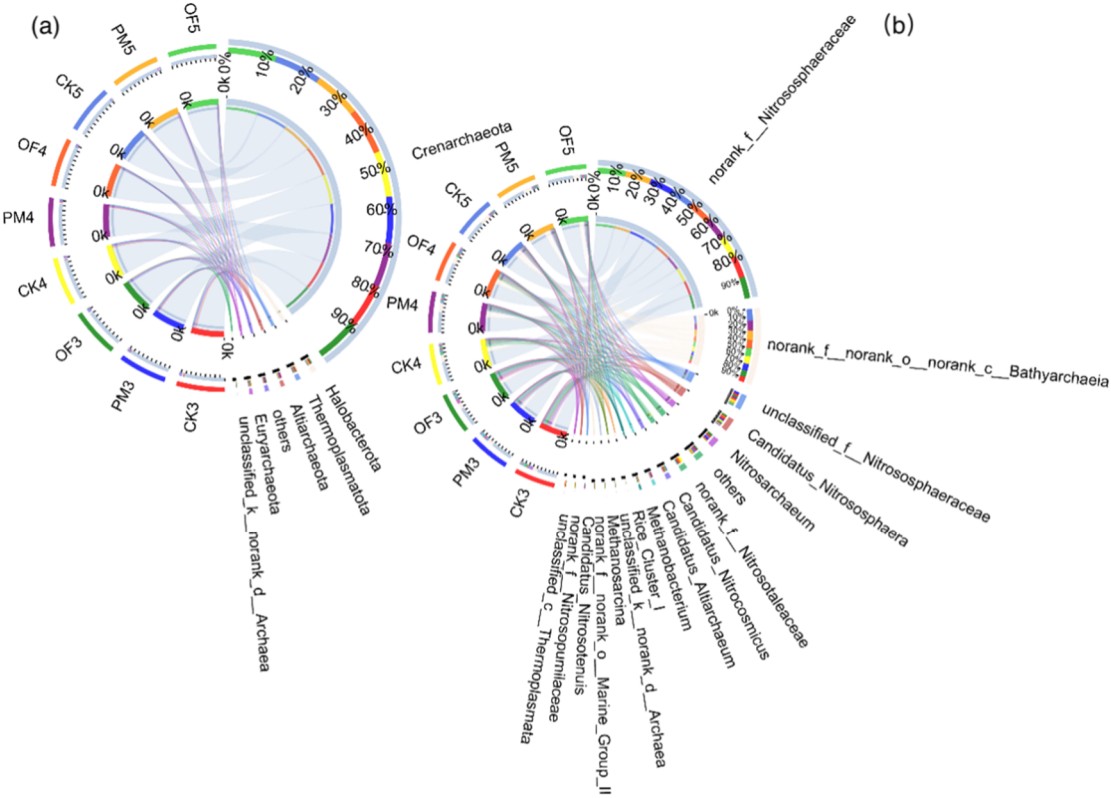

**Figure 2.** Composition and relative abundance of microbial phylum and genus levels in paddy field soil: (**a**) phylum level; (**b**) genus level. (CK3, PM3, OF3—Heading stage; CK4, PM4, OF4—Flowering stage; CK5, PM5, OF5—Mature stage).

At the genus level, the relative abundance of *Candidatus_Nitrososphaera* in different treatment groups decreased gradually with the growth of rice, with the CK and OF groups being the highest at the heading stage, accounting for 14.0%, and the PM group being the lowest at 12.0%. At the flowering stage, the ranking was OF > CK > PM; at the mature stage, the differences among the three groups were small, and the OF group was still the highest, accounting for 9.7%. At heading stage, the relative abundance of *Nitrosarchaeum* in the OF group was the highest, accounting for 23%, and was 1.0% and 9.0% higher than that in the CK and PM groups; at flowering stage, the ranking was PM > OF > CK, and the PM group was 5.7% and 2.9% higher than that in the CK and OF groups; finally, at the mature stage, the percentage of the CK group increased to 15.0%, and that of the PM and OF groups decreased to 2.6% and 1.9%, respectively. Manure application increased the relative abundance of *Candidatus_Nitrocosmicus*, which was particularly significant at the mature stage, with 6.30% and 5.30% higher in the OF and PM groups than in the CK group, respectively. *Candidatus_Nitrocosmicus* functions as a nitrogen cycling subclass, and the increase in its relative abundance was beneficial to the nitrogen cycling in paddy soils.

At the heading stage, the relative abundance of *Methanobacterium* in the OF group was the highest at 15.0%, which was 5.1% and 7.2% higher than that in the CK and PM groups; at the flowering stage, the CK group increased and the OF group decreased, with the PM

group being the highest at 16.0%, followed by the CK group at 12.0% and the OF group at the lowest, at 5.8%. Finally, at the mature stage, the PM group was still the highest at 16.0%, followed by the OF group at 11.0%, and the lowest at 6.2% in the CK group. At the heading stage, the relative abundance of *Methanosarcina* in the OF group was higher at 41.0%, while the CK and PM groups accounted for 14.0% and 7.0%, respectively; at the flowering stage, the relative abundance of *Methanosarcina* in the OF group decreased sharply, with small differences among the three groups, 3.7% in the CK group, 7.6% in the PM group, and 3.0% in the OF group. Finally, at the mature stage, there was an increase in the PM and OF groups, 12.0% and 8.1%, respectively, while the CK group remained at 3.7%.

The results of similarity or difference analysis of archaeal community composition under different manure treatments are shown in Figure 3. Manure application changed the community composition of soil Archaea in rice fields; Fu et al. also had similar results [20], which were especially noticeable at the heading and mature stage. At the heading stage (a), there were some differences in archaeal community composition between different treatment groups. The CK group was distributed in both positive and negative half-axes of PC1 and PC2, while the PM group was distributed in both positive and negative half-axes of PC1, mainly in the negative half-axes of PC2, and the OF group was mainly distributed in positive half-axes of PC1 and negative half-axes of PC2. At the flowering stage (b), the differences in archaeal community composition between treatment groups were small; the CK group was distributed in both positive and negative half-axes of PC1 and PC2, the PM group was mainly distributed in the negative half-axes of PC1 and PC2, while the OF group was mainly distributed in the positive half-axes of PC1 and PC2. Finally, at the mature stage (c), the archaeal community composition of the fertilized groups was similar and differed from that of the CK group, with the CK group mainly distributed in the negative half-axis of the PC1 and PC2, the PM group mainly distributed in the positive half-axis of PC1 and in the positive and negative half-axis of PC2, while the OF group was distributed in the positive half-axis of the PC1 and PC2.

A one-way ANOVA was selected to analyze the differences in microbial composition between the different treatment groups based on the relative abundance of microorganisms at the phylum and genus levels (Figure 4). No significant differences in the relative abundance of Archaea were found between the different treatment groups at the phylum level at the heading and flowering stage (a, b). At the genus level, the relative abundance of *Methanobrevibacter* was significantly higher in the PM group than in the OF and CK groups at the heading stage ($p < 0.01$), 32.83-fold and 100.51-fold higher, respectively. In the flowering stage (b), the relative abundance of *Methanobrevibacter* in the PM and OF groups was significantly higher than that in the CK group ($p < 0.01$), 34.30-fold and 15.20-fold higher, respectively. The study of Yuan et al. [21] showed that the relative abundance of *Methanobrevibacter* increased in the soil of paddy fields with compost application compared to the control group, which is consistent with the results of this study; meanwhile, the relative abundance of *Methanosphaera* in the PM group was significantly higher than that in the OF and CK groups ($p < 0.05$), both 7.00-fold higher.

The relative abundance of *Euryarchaeota* was highest in the PM group at the phylum level at mature stage (c), which was higher than that of the CK group by 1.60-fold ($p < 0.05$). The study of Brauer et al. [22] showed that *Euryarchaeota*, as a common methanogen in soil, is essential for soil methane emission processes, and manure application increased its relative abundance, which may lead to an increase in $CH_4$ emissions. Pig manure and organic fertilizer application significantly reduced the relative abundance of *Altiarchaeota* ($p < 0.05$), *Nanoarchaeota* ($p < 0.05$), *Asgardarchaeota* ($p < 0.05$), *lainarchaeota* ($p < 0.05$), and *Micrarchaeota* ($p < 0.01$). At the genus level, the relative abundance of *norank_f__Nitrososphaeraceae* in the OF group was the highest, significantly higher than that in the CK group ($p < 0.05$), 21.72% higher; *norank_f__Nitrososphaeraceae* is an ammonia-oxidizing microorganism, and the increase in the relative abundance of manure after application may facilitate the nitrification process in the soil. The relative abundance of *Methanobacterium* in the PM group was significantly higher than that of the CK group ($p < 0.05$), 157.05% higher, which is consistent

with the research results of Fu et al. [20]. The relative abundance of *Methanobrevibacter* in the PM group was significantly higher than that in OF and CK groups, 36.33 and 21.40 times higher, respectively (*p* < 0.01).

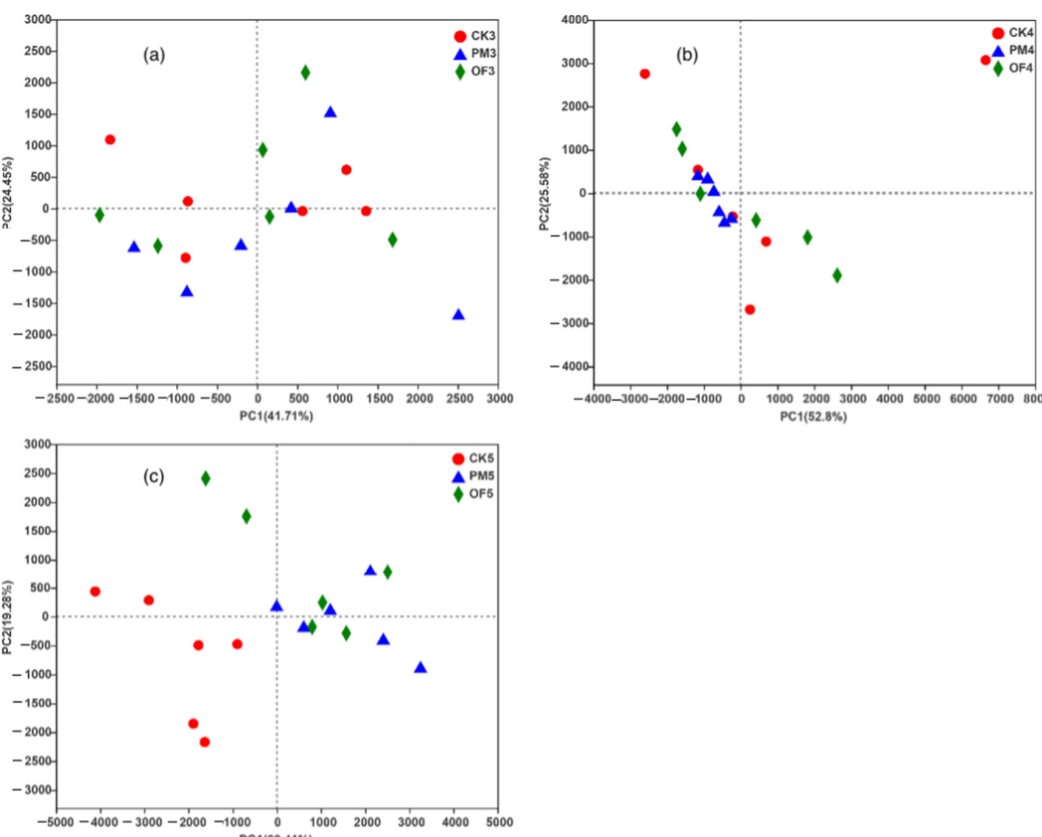

**Figure 3.** Redundant analysis of archaeal community phylum level: (**a**) Heading stage; (**b**) Flowering stage; (**c**) Mature stage.

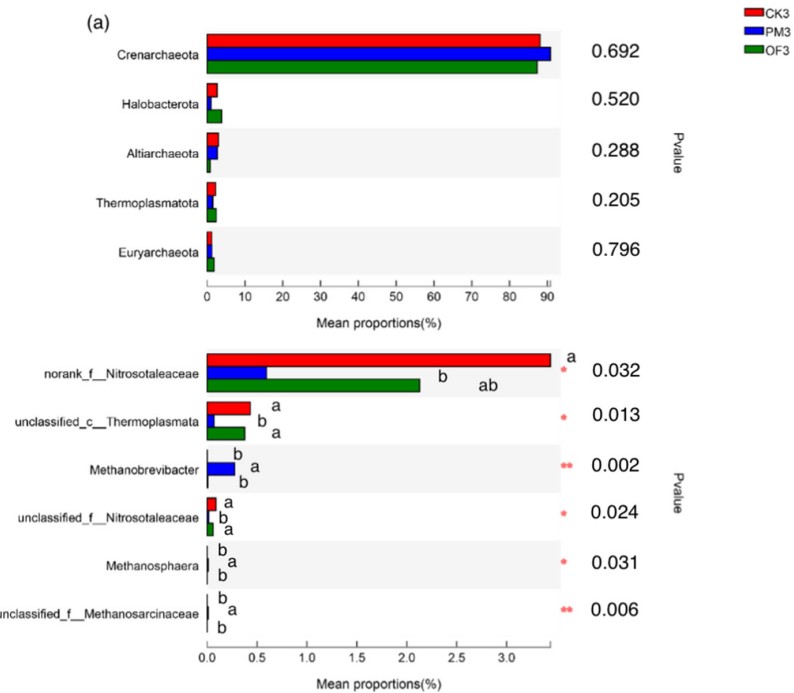

**Figure 4.** *Cont.*

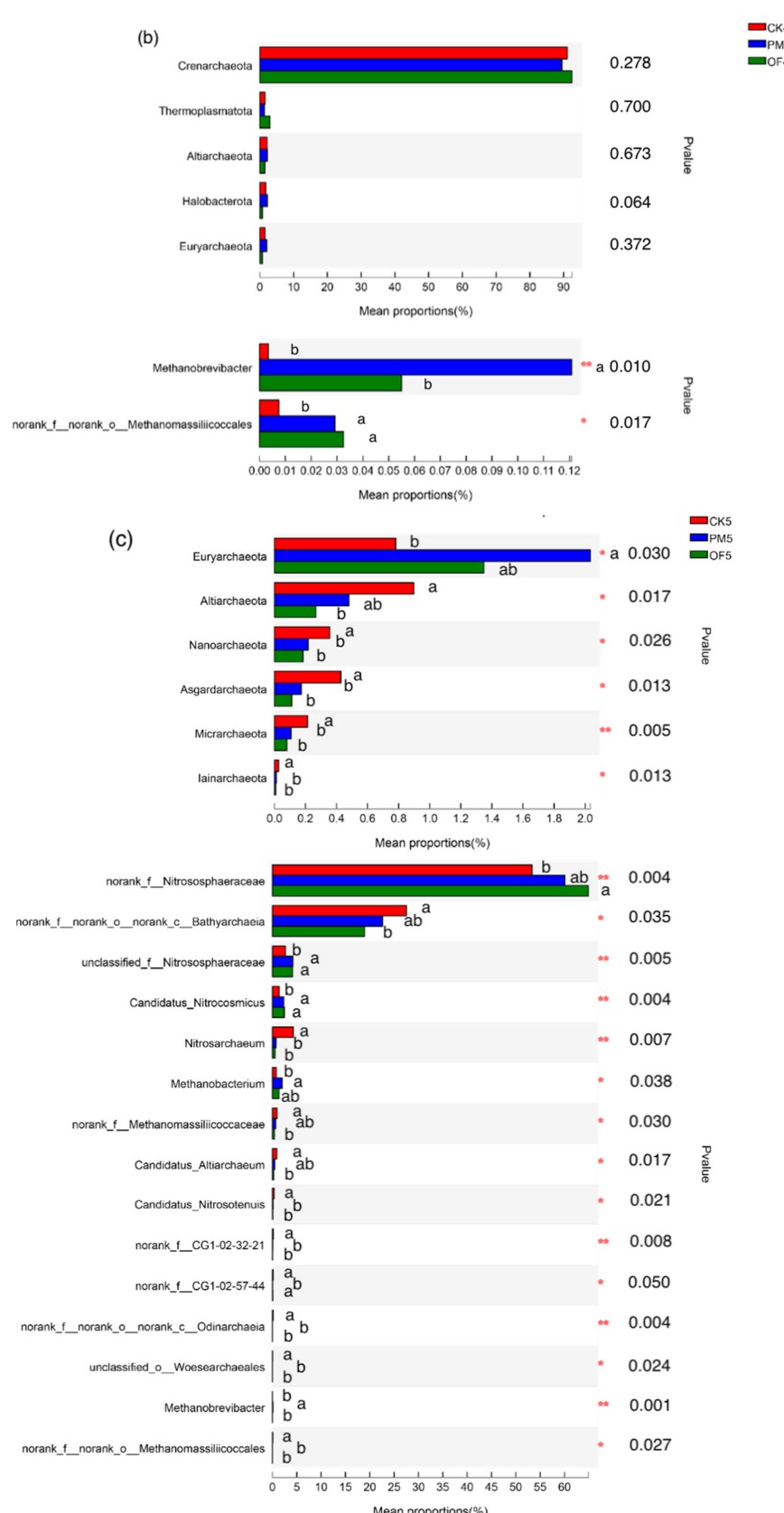

**Figure 4.** Univariate ANOVA of intergroup differences in microphylum and genus levels in paddy fields: (**a**) Heading stage; (**b**) Flowering stage; (**c**) Mature stage. * indicates ($p < 0.05$), ** indicates ($p < 0.01$).

*3.4. Effect of Manure Application on $CH_4$ Emission from Paddy Soils*

The effects of pig manure and organic fertilizer application on soil $CH_4$ emissions in rice fields are shown in Figure 5. At the heading stage, the highest $CH_4$ emissions peaked in the PM group, and $CH_4$ emissions in PM and OF groups were 233.23% and 176.93% higher than those in the CK group, reaching 7.611 nmol·$(m^2·s)^{-1}$ and 6.325 nmol·$(m^2·s)^{-1}$, respectively, while there was no significant difference between the three groups. At the flowering stage, the lowest $CH_4$ emissions were reached in the PM group, at −1.948 nmol·$(m^2·s)^{-1}$, and $CH_4$ emissions from all three treatment groups decreased, among which the PM group decreased more noticeably by 9.559 nmol·$(m^2·s)^{-1}$, the OF group was significantly higher than the pig manure group, higher than 5.971 nmol·$(m^2·s)^{-1}$ ($p < 0.05$). At the mature stage, $CH_4$ emissions from all three groups stabilized: all three groups tend to be 0. At the mature stage, $CH_4$ was measured after drainage and drying in the rice field, which is the main reason for $CH_4$ emissions of different treatment groups to be 0. The results indicated that the application of pig manure and organic fertilizer less affected soil $CH_4$ emissions in rice fields. With the growth of rice, the $CH_4$ emissions of all three groups showed a decreasing trend. In conclusion, the rational application of fresh pig manure and organic fertilizer did not significantly increase soil methane emission fluxes in paddy fields, and thus had less impact on the greenhouse effect.

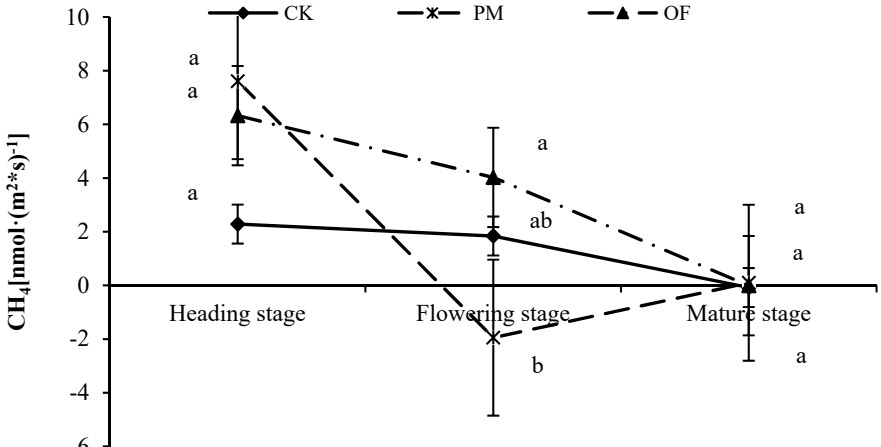

**Figure 5.** The effect of manure application on $CH_4$ emission from paddy field soil. Different lowercase letters indicate significant differences between groups ($p < 0.05$).

## 4. Discussion

*4.1. Effect of Manure Application on Soil Archaeal Microorganisms in Paddy Fields*

The phyla with relatively high abundance at the phylum level after manure application were *Crenarchaeota*, *Halobacterota*, *Thermoplasmatota*, *Altiarchaeota*, and *Euryarchaeota*, and these microorganisms may play an important role in the emission of soil $CH_4$ from rice fields. Manure application increased their relative abundance to different degrees, and the study of Li et al. [23] found that *Crenarchaeota* may play a crucial role in soil ammonia oxidation after studying the archaeal community composition of heavy metal-contaminated soils in mining areas. A study by Brauer et al. [22] showed that *Halobacterota*, *Thermoplasmatota* and *Euryarchaeota,* as common methanogens in soils, are essential for soil methane emission processes, according to metagenomic data, as at least a few organisms within this group contain the essential gene, mcrA.

Regarding *Candidatus_Nitrososphaera*, as a microorganism associated with nitrogen cycling and oxygen release [24], fertilization increased its relative abundance and may facilitate the soil nitrogen cycling processes in paddy fields. Meanwhile, *Candidatus_Nitrososphaera* can act as an abundant ammoxidation catalyst and degradability agent of heterologous compounds [25], while the increase in its relative abundance after applying the organic fertilizers promoted soil nitrogen cycling processes as well as microbial growth. The

application of pig manure and organic fertilizer increased the relative abundance of *Candidatus_Nitrocosmicus*, an archaeon with nitrification, which may play an important role in the nitrogen cycle [26]. Most methanogens (such as *Methanobacterium*) use acetate as methanogenic substrate, while a few of them can use $H_2/CO_2$ as substrate, and they are mainly mesophilic or thermophilic Archaea that can grow and develop in a weak alkaline environment. After the soil organic matter enters the anoxic environment, it is generally decomposed through the following steps: (1) hydrolysis, the complex organic matter is converted into monosaccharide substances, and further fermented into fatty acids, $CO_2$ and $H_2$; (2) under the action of syntrophs, acetic acid, $CO_2$ and $H_2$ are generated; and (3) acetic acid, $H_2$ and $CO_2$ are respectively utilized by acetate-type and hydrogen-type methanogenic Archaea to produce methane. Homoacetogenic bacteria can also convert $H_2/CO_2$ to acetic acid and acetogenic methanogenic Archaea to methane. The anaerobic degradation of organic matter is coordinated by the above-mentioned anaerobic food chain, methane is the final product of the anaerobic food chain, and methanogens are the terminal members of the food chain for anaerobic degradation of organic matter [27]. Organic fertilizer application increased the relative abundance of *Methanosarcina*, which may be related to the metabolic capacity of methane and its derived compounds accumulated from the decomposition of organic matter introduced in organic agriculture. Wen et al. [25] found that *Methanosarcina*, *Methanosaeta* and *Methanobrevibacter* may contribute to the reduction of $CH_4$ emissions after studying the effect of high temperature compost on methanogenic microbial communities; high-temperature composting affected $CH_4$-related microbial communities at 80 °C, where the abundance of most methanogens and methanotrophs was suppressed. At the heading stage, the relative abundance of *Rice_Cluster_I* was lower in both PM and OF groups than in the CK group, reaching 14.3% and 6.0%, respectively; at flowering stage, the PM group increased and was 1.0% higher than the CK group, while the OF group remained the lowest. Finally, at the mature stage, the PM group remained the highest, accounting for 11.0%, while the OF and CK groups were less different, at 6.6% and 7.1%, respectively. Studies based on 16S rRNA and mcrA gene show that *Rice_Cluster_I* is widely distributed in the environment and spread across different terrestrial and coastal ecosystems around the world. Paddy fields around the world are one of the main habitats of *Rice_Cluster_I* [27]. Taxonomic characterization based on 16S rRNA gene sequencing by Valenzuela et al. [28] showed that *Rice_Cluster_I* and uncultured Archaea of the methanogenic family are microorganisms that may be involved in anaerobic oxidation of methane (AOM), and these microorganisms are associated with humus-mediated $N_2O$ reduction. Studies based on stable isotopic probe technology (SIP) have confirmed that *Rice_Cluster_I* in the rice rhizosphere plays a key role in the formation of methane [29].

*4.2. Effects of Changes in Environmental Factors on Soil Archaeal Microorganisms in Paddy Fields*

Microorganisms constantly change the environment and adapt to the environment under the influence of the changing environment. Different environmental factors affect the abundance and community structure of Archaea in paddy soil (Figure 6). The application of manure and its organic fertilizers can change the soil environment, then affecting the abundance, community structure and diversity of Archaea, and ultimately affecting the $CH_4$ emission flux. A study by Yuan et al. [21] showed that there is a significant effect of soil environmental factors on the community structure of methanogens in paddy soils. In this paper, we analyzed the degree of influence of environmental factors in different treatment groups on the horizontal community composition of soil Archaea in paddy fields by Spearman correlation. The results showed that *Methanomassiliicoccus* was significantly positively correlated *(p < 0.05)* with TN, NN (ammonia nitrogen) ($p < 0.01$) and Cr, and negatively correlated with Zn ($p < 0.001$) and T ($p < 0.01$) in the CK group. *Candidatus_Methanomethylicus* was significantly positively correlated with TN, NN ($p < 0.05$) and MC ($p < 0.01$), and negatively correlated with Zn ($p < 0.05$). *Methanoregula* was significantly and negatively correlated with Cu and Ni ($p < 0.05$), *Methanosaeta* was significantly and negatively correlated with Cu ($p < 0.05$), *Nitrosarchaeum* was significantly and positively

correlated with Pb, AN ($p < 0.05$), TN ($p < 0.01$), OM and TK ($p < 0.001$). *Methanolobus* was significantly and positively correlated with TK ($p < 0.05$). *Methanosarcina* was significantly and positively correlated with OM and TK ($p < 0.05$).

*Methanobrevibacter* was significantly and positively correlated with EC, Pb *($p < 0.05$)*, Cr ($p < 0.01$), pH and Ni ($p < 0.001$) in the PM group. *Methanosphaera* was significantly and positively correlated with EC ($p < 0.05$). *Methanolobus* was significantly and positively correlated with MC ($p < 0.05$). Pampillon-Gonzalez et al. [30] used metagenome techniques to study pig fecal bacteria and archaeal communities, and found that *Methanosarcina*, *Methanolobus*, *Methanosaeta* and *Methanospirillum* were some of the more abundant Archaea, which is consistent with the results of this study. *Methanoregula* was significantly and negatively correlated with TK, pH, Ni, TN ($p < 0.05$) and Cr ($p < 0.01$). The study of Liu et al. [31] showed that urea application stimulated the *Methanoregula* in paddy soils and changed oxygen state of the paddy soil and water; while, in this paper, fresh pig manure application changed soil physicochemical properties, which maybe the main reason for the significant negative correlation of *Methanoregula* with TK, pH, Ni, TN and Cr. *Methanolinea* was significantly and negatively correlated with TK ($p < 0.05$), Cr, Ni, TN ($p < 0.01$) and pH ($p < 0.001$). Taxonomic characterization based on 16S rRNA gene sequencing showed that *Rice_Cluster_I* may be involved in anaerobic oxidation of methane, which is associated with humus-mediated $N_2O$ reduction, and this may be the reason for its significant negative correlation with TN. The study of Conrad et al. [32] showed that most of the $CH_4$ emitted from rice fields comes from its photosynthesis, in which the photosynthetic products produced are converted to $CH_4$, and most of the $CH_4$ emitted from rice fields comes from plant photosynthesis, in which the photosynthetic products will be transformed into $CH_4$; conversely, in rice rhizosphere, *Rice_Cluster_I* is the methanogenic Archaea that causes this phenomenon. $H^+$ is produced during photosynthesis, resulting in the decrease of pH, which may be the main reason for its significant negative correlation with pH. The study of Fey et al. [33] showed that *Rice_Cluster_I* was the most common methanogenic group in paddy soils, and its relative abundance was negatively correlated with temperature. *Methanocella* was significantly and negatively correlated with pH, Ni, TN ($p < 0.05$) and Cr ($p < 0.01$).

*Methanogenium* was significantly and positively correlated with NN ($p < 0.05$) and negatively correlated with Pb, T ($p < 0.05$) and TN ($p < 0.01$) in the OF group. Kudo et al. [34] analyzed methanogens in soil samples from nine rice fields in Japan and showed that *Methanogenium* was its major methanogen. *Methanomassiliicoccus* was significantly and positively correlated with TK, Cr ($p < 0.05$), Hg and Ni ($p < 0.01$). Study by Jang et al. [35] showed that adding biochar to anaerobic digestion of manure may have stimulated the growth of *Methanomassiliicoccus*, while low concentrations of heavy metals such as Cr, Hg, and Ni in organic fertilizers may stimulate the growth and reproduction of *Methanomassiliicoccus* to varying degrees. *Rice_Cluster_I* was significantly and positively correlated with Cr and Ni ($p < 0.001$). *Methanocella* was significantly and positively correlated with As ($p < 0.05$), Cr and Ni ($p < 0.01$). Wang et al. [36], in their research on the composition of nitrogen-fixing bacterial communities and their symbiotic patterns in the soil profiles of paddy fields of three soil types in China, found that *Methylomonas*, *Methanocella* and *Methanosaet* were the main Archaea of the surface soil (0–40 cm). The study by Yuan et al. [21] showed that, after applying the different type of fertilizers and detecting the methane emissions and methanogens in the rhizosphere found that *Methanocella* was its main methanogens in the paddy field. *Methanosarcina* showed a significant positive correlation with As, Cu ($p < 0.05$) and Ni ($p < 0.01$). The study of Wang et al. [37] showed that the dominant methanogens in Ni- and Co-contaminated paddy soils were *Methanobacterium*, *Methanosarcina*, *Methanocella*, *Methanomassiliicoccus*, and *Rice_Cluster_I*, and that the addition of heavy metals with low concentration in paddy soils may be the leading reason for promoting the activity of methanogens. The application of pig manure and organic fertilizer brought different contents of heavy metals into the paddy soil, which may be the reason for the high abundance of methanogens in PM and OF groups.

*Methanolobus* was significantly and positively correlated with Ni ($p < 0.05$); *Methanobacterium* was significantly and positively correlated with As ($p < 0.05$); and *Methanospirillum* was significantly and negatively correlated with Hg ($p < 0.05$). The study by Zhang et al. [38] showed that *Methanospirillum* was positively correlated with methane production capacity ($p < 0.05$), and they play a potential contribution in methanogenesis, while a high concentration of Hg inhibited the growth and reproduction of methanogens, which may be the main reason for its significant negative correlation with Hg. *Methanobrevibacter* was significantly negatively correlated with AN and MC ($p < 0.01$), but positively correlated with TN, Zn ($p < 0.05$) and T ($p < 0.01$). *Nitrosarchaeum* was significantly and positively correlated with Cu, pH ($p < 0.05$), TK ($p < 0.01$), T, TN, and Zn ($p < 0.001$), and negatively correlated with Cd ($p < 0.05$) and MC ($p < 0.001$). *Nitrosarchaeum* is involved in the process of soil nitrogen cycling in paddy fields, and high concentrations of TK and TN in organic fertilizers can provide sufficient nutrients to promote its growth and reproduction.

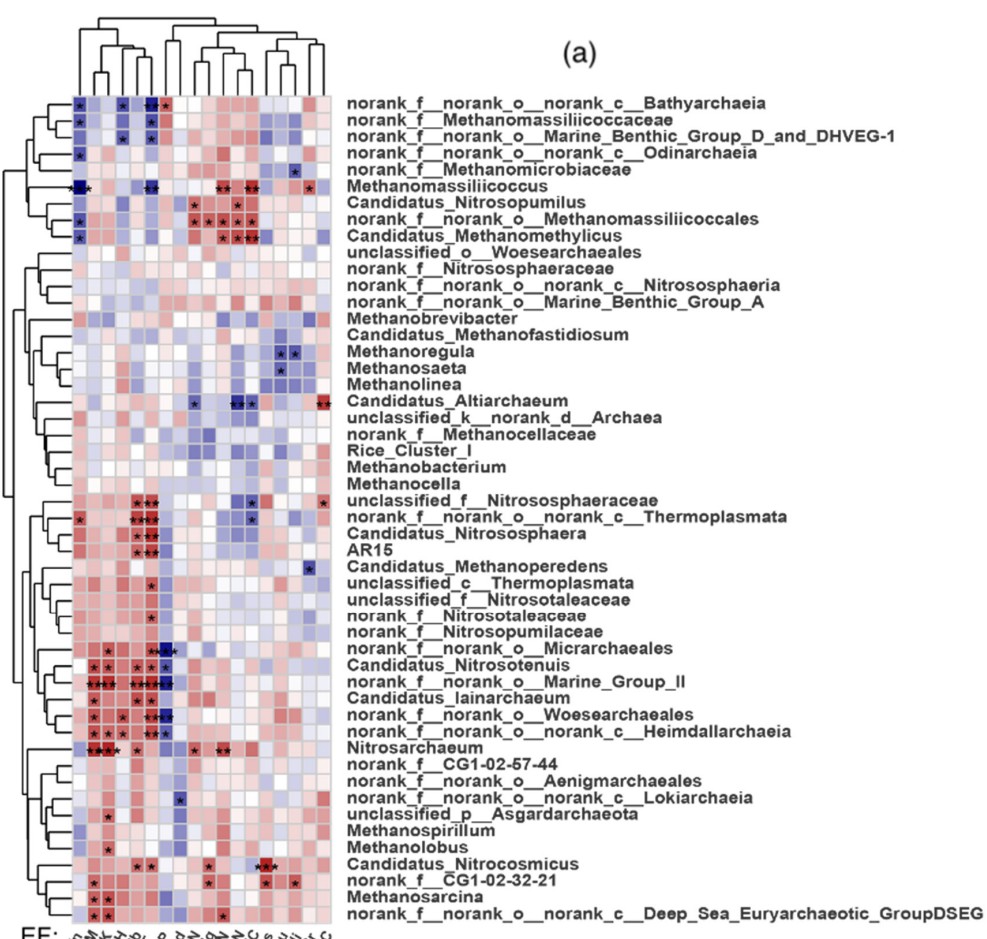

**Figure 6.** *Cont.*

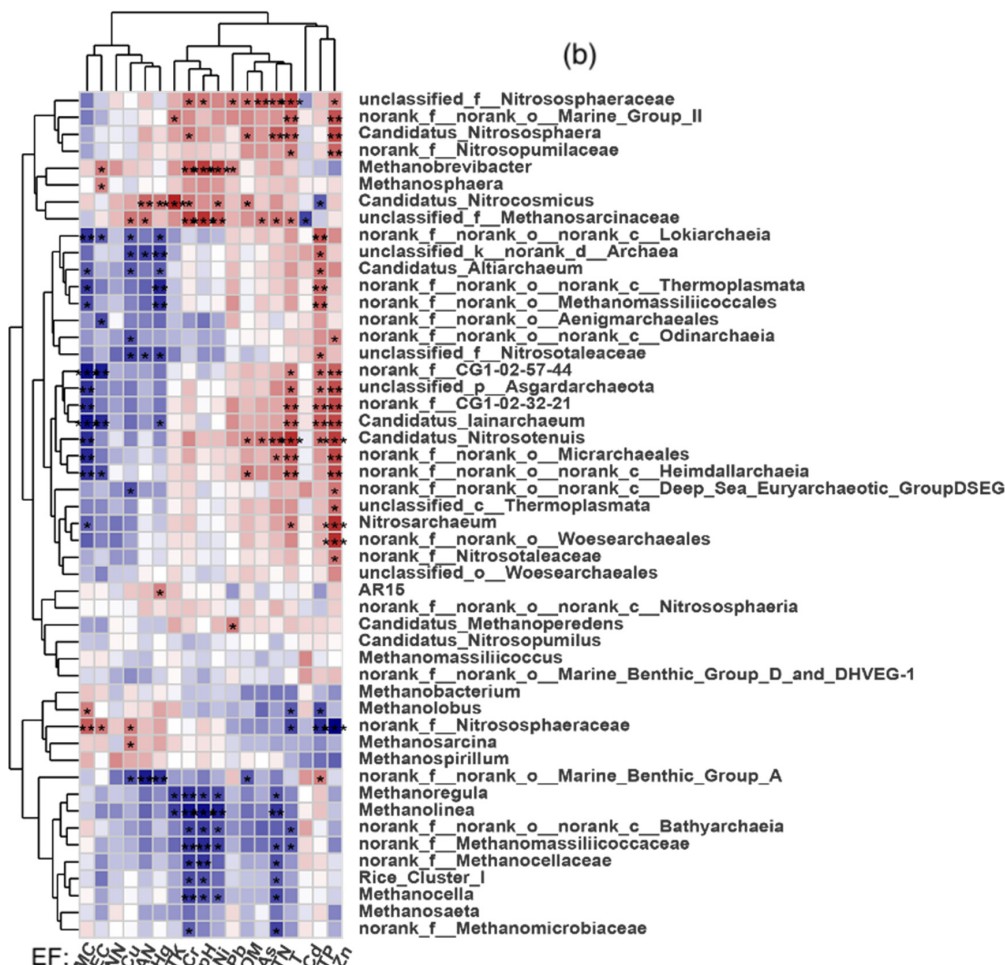

**Figure 6.** *Cont.*

## Spearman Correlation Heatmap

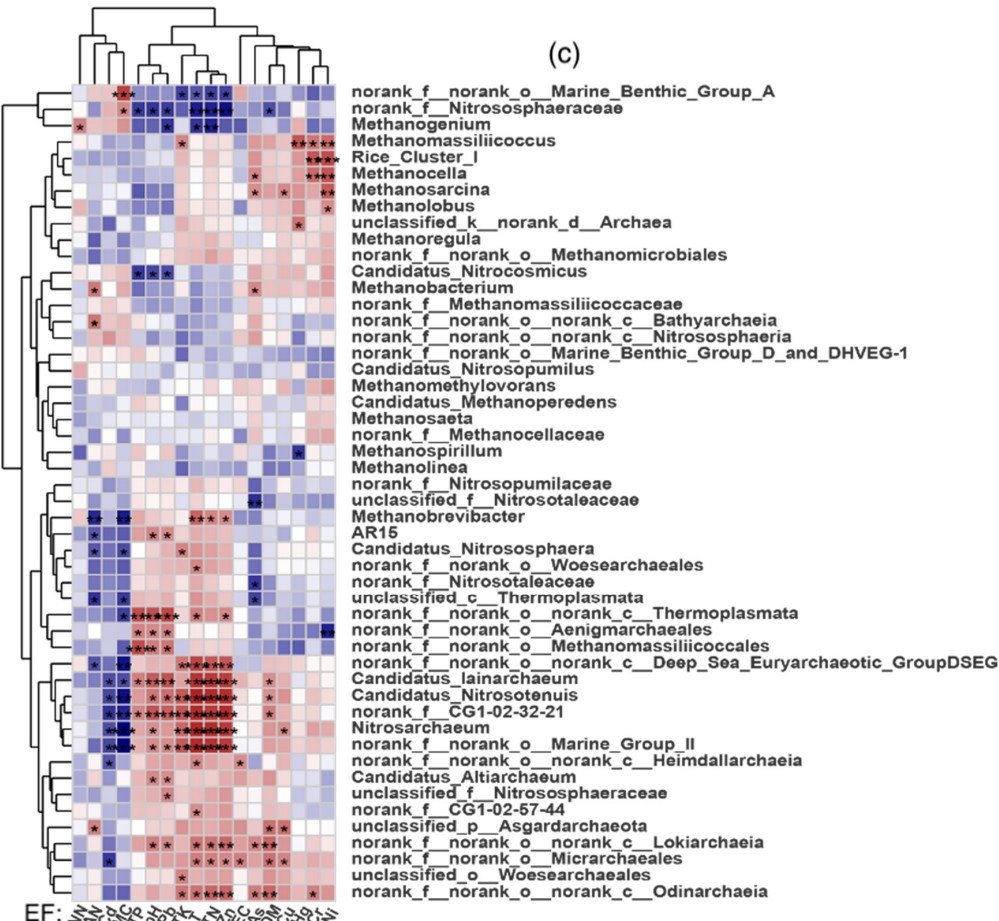

**Figure 6.** Heatmap of the correlation between environmental factors and microflora in different treatment groups: (**a**) CK group; (**b**) PM group; (**c**) OF group. EF, environmental factor. * Indicates ($p < 0.05$), ** indicates ($p < 0.01$), *** indicates ($p < 0.001$).

TN, NN, TK and OM were the main environmental factors affecting the composition of methanogen communities in paddy soils in the CK group. In the PM group, EC, Cr, pH, Ni, TK and TN were the main environmental factors, while in the OF group, TN, TK, Cr, Hg, Ni and As were the main environmental factors. The environmental factors that affect paddy Archaea in the CK group were mainly nutrient elements, while heavy metals, N and K were the main factors in the pig manure and organic fertilizer groups, which may be the manure contains the heavy metals and nitrogen and potassium elements, introduced into the paddy soil, and affected its microorganisms. All treatment groups showed a correlation between environmental factors and archaeal community changes, which may be due to the fact that different manure fertilizers changed the soil environmental factors in the paddy field, and the changes in environmental factors affected the soil archaeal community in the paddy field, while the changes in archaeal community structure in turn affected the soil environmental factors, which is a complex process of mutual influence and interaction between the two.

### 4.3. Effects of Changes in Methanogens on CH$_4$ Emission in Rice Fields after Manure Application

Studies have shown that soil environment, water and fertilizer management, and rice reproductive period can cause changes in the community structure of methanogens in paddy fields, and thus affect CH$_4$ emissions from paddy fields [39–42], with fertilization measures having a particularly significant effect. Manure application increased the emission

flux of $CH_4$ from paddy soils, and there was a significant difference in the flowering stage, the OF group was significantly higher than the PM group ($p < 0.05$). The PM group had the highest $CH_4$ emission flux of 7.611 nmol·$(m^2·s)^{-1}$ at the heading stage, followed by the OF group with 6.325 nmol·$(m^2·s)^{-1}$, while the CK group had the lowest. At the heading stage, the relative abundance of *Methanobacterium* was highest in the OF group; the relative abundance of *Methanobrevibacter* was significantly higher in the PM group than in the OF and CK groups ($p < 0.01$); the relative abundance of *Methanobacterium* and *Methanobrevibacter* increased significantly in paddy soil after the application of pig manure and its organic fertilizer. The relative abundance of *Methanosarcina* was higher in the OF group; Pampillon-Gonzalez et al. [30] used macrogenomics techniques to study the pig manure archaeal community and found that *Methanosarcina* was among the more abundant Archaea. The $CH_4$ emission flux at the flowering stage was higher in the OF group than in the CK and PM groups, by 2.183 nmol·$(m^2·s)^{-1}$ and 5.971 nmol·$(m^2·s)^{-1}$, respectively. The relative abundance of *Methanobacterium* in the PM group was the highest, followed by the CK group, and was the lowest in the OF group. The relative abundance of *Methanobrevibacter* in the PM and OF groups was significantly higher than that in the CK group ($p < 0.01$); meanwhile, the relative abundance of *Methanosphaera* in the PM group was significantly higher than that in the OF and CK groups ($p < 0.05$). The abundance of methanogens in the PM group was higher, but the $CH_4$ emission flux was the lowest. The main reason may be that there was ponding in PVC pipes during the measurement of methane emission flux in the PM group. The ponding and microorganisms in the ponding led to the negative measurement of $CH_4$ emission flux. The study of He et al. [43] showed that *Methanobrevibacter* and *Methanosarcina* accounted for more than 80.0% of the methanogens in pig manure and organic fertilizer made from aerobic composting of pig manure and straw. This is a good explanation as to why the relative abundance of *Methanosarcina* increased after pig manure and organic fertilizer were applied to the soil. The study of Fu et al. [20] showed that the archaeal community structure changed noticeably after the addition of rice straw, manure or wood chip biochar to the rice soils, and *Methanobacterium* and *Methanosarcina* were noticeably enriched in the different treatments. Finally, rice was harvested at maturity, resulting in $CH_4$ emission fluxes tending to 0 for all three treatment groups, but the relative abundance of methanogens in the soil of the fertilized group was still higher than that of the CK group. The methane emission flux of different treatment groups tends to 0 in the mature stage, the reason for which is the measurement of methane emission flux after rice field drainage and drying.

There was an increase in the relative abundance of *Methanobacterium*, *Methanobrevibacter*, *Methanosphaera*, *Methanosarcina* and *Rice_Cluster_I* after applying the manure, which is probably the main reason for the increase in $CH_4$ emission flux in the PM and OF groups. Manure fertilizers increased the organic matter, TN and TK content of paddy soils, which could provide sufficient nutrients for methanogens, thus promoting the growth and reproduction of methanogens in paddy fields, which could be the main reason for the increase in $CH_4$ emission flux in the PM and OF groups. A study by Yuan and others found that the increase in $CH_4$ flux emissions from paddy soils after organic fertilizer application was mainly from native soil and photosynthesis, and that $CH_4$ emissions and unstable methanogenic community structure were mainly due to increases in soil carbon, nitrogen, potassium, phosphate, and acetate [21]. These findings highlight the contribution of native soil and photosynthesis-derived carbon in $CH_4$ emissions from paddy soils, and provide a basis for studies of the pathways involved in more complex ecosystem $CH_4$ processes.

## 5. Conclusions

(1)  Compared with the control group, manure application increased the contents of Zn, Cr, and Ni in the paddy soil, but none of them were significantly different, and their contents gradually decreased with the growth of rice. Manure application increased the content of soil organic matter, TP and TK, which provided sufficient nutrients for rice growth.

(2) Compared with the control group, manure application reduced the diversity and abundance of soil Archaea in paddy fields. TN, NN, TK and OM were the main environmental factors affecting the composition of soil methanogenic communities in the CK group; that in the PM group were EC, Cr, pH, Ni, TK and TN; and that in the OF group were TN, TK, Cr, Hg, Ni and As.

(3) *Methanobacterium*, *Methanobrevibacter*, *Methanosphaera*, *Methanosarcina* and *Rice_Cluster_I* were the main methanogens in paddy soils after manure application, and the increase in the relative abundance of methanogens was probably the main reason for the increase in $CH_4$ emission flux. Meanwhile, these Archaea are also involved in the carbon cycle of the paddy soil ecosystem.

**Author Contributions:** Conceptualization, J.W.; Data curation, P.L., L.S. and M.M.; Formal analysis, J.W. and C.Y.; Funding acquisition, J.W.; Investigation, P.L., B.X. and S.Z.; Methodology, J.W., M.W. and L.X.; Project administration, B.X., S.Z. and C.S.; Resources, J.W., M.W. and C.S.; Software, P.L., L.S. and M.M.; Supervision, S.H.; Validation, J.W. and M.W.; Visualization, J.W., C.Y. and L.X.; Writing—original draft, J.W.; Writing—review & editing, M.W. and S.H. All authors have read and agreed to the published version of the manuscript.

**Funding:** This study was supported by the Science and Technology Commission Foundation of Shanghai (No. 20dz1204800); the National Natural Science Foundation of China (No. 51979168); and the Natural Science Foundation of Shanghai, China (No. 19ZR1443900, 19ZR1444100).

**Institutional Review Board Statement:** Not applicable.

**Informed Consent Statement:** Not applicable.

**Data Availability Statement:** Not applicable.

**Conflicts of Interest:** The authors declare no conflict of interest.

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
