# Peer review of "Effects of Pig Manure and Its Organic Fertilizer Application on Archaea and Methane Emission in Paddy Fields"

_land, doi:10.3390/land11040499_

Round 1
Reviewer 1 Report
Land-1631859
Effects of pig manure and its organic fertilizer application on methanogens and methane emission in paddy fields.
Wu et al.
Review of revised submission
Overall
These authors have considerably improved their manuscript with revisions since the last version I reviewed. However, the use of English is still not up to the high standards of the journal, and some arguments and conclusions are poorly supported. I am gratified to see the addition of clarity to most sections, and the addition of uncertainty in most statements about the causes of various differences found in the Results.
Abstract
LN27 – the English name for the element with atomic number 19 is Potassium, not Kalium as it is in some other languages.
Methods
LN142-144 – was TP measured two different ways?
Results
The paragraph from LN219 to LN235 compares some of the results of the current study with other studies – this is text more appropriate for the Discussion section.
Discussion
LN500 – is "Fey, A" a citation? Does it have a number?
What do the black stars on some heatmap positions indicate in Figure 6?
LN586 – how would ponding affect methane flux? Please explain. Did the ponding cause a measurement error?
Author Response
Dear reviewer:
Thank you very much for appreciating our work and for your kind suggestions for the improvement of our manuscript. The revisions have been amended in the revised manuscript (in red). Kindly consider the changes as follows.
Comment 1: Abstract
LN27 – the English name for the element with atomic number 19 is Potassium, not Kalium as it is in some other languages.
Response 1: Thank you for your good comment. Have been revised (Line 27).
Comment 2: Methods
LN142-144 – was TP measured two different ways?
Response 2: Thank you for your good comment. Have been revised (Line 145).
Comment 3: Results
The paragraph from LN219 to LN235 compares some of the results of the current study with other studies – this is text more appropriate for the Discussion section.
Response 3: Thank you for your good comment. The discussion content in the manuscript results helps readers understand the research content of the article more easily.
Comment 4: Discussion
LN500 – is "Fey, A" a citation? Does it have a number?
Response 4: Thank you for your good comment. Have been revised (Line 534).
Comment 5: What do the black stars on some heatmap positions indicate in Figure 6?
Response 5: Thank you for your good comment. Have been revised (Lines 593-594). *Indicates (P < 0.05), * * indicates (P < 0.01), * * * indicates (P < 0.001).
Comment 6: LN586 – how would ponding affect methane flux? Please explain. Did the ponding cause a measurement error?
Response 6: Thank you for your good comment. The methane emission flux is negative because the water in the PVC ring has a great impact on the change of gas concentration, resulting in the inability of methane to be discharged and absorbed by the soil.
Reviewer 2 Report
The work assesses the effect of manure on methane and methane emissions in rice fields.
Three types of fertilization were investigated in the field experiment
controls (without fertilizer), pig manure and organic fertilizer made from pig manure. The effect of manure application on CH4 and soil microorganisms by the use of microbial DNA extracted from soil samples was assessed in rice at heading, flowering and maturity.
Manuscript follows the layout of a scientific article. It is written in plain English with minor flaws. The analysis of soil samples and the measurement of the emitted methane are described very carefully.
Statistical analyses of the comparison of soil physicochemical factors and the diversity of the Archaea are described. However, it does not include cluster analysis (if performed for Figure 2), nor principal component analysis. For microbial analysis, is the one-way ANOVA a good test, does the data meet the assumption of normally distributed?
The Results section begins with a description of the heavy metal content of the soil after manure has been applied. This is not the main purpose of the work. If it is to remain in this form, it should be mentioned in Introduction that it was one the purpose of this work or how it can influence the measured variables.
What is Figure 2 based on? Is this the result of a cluster analysis? In my opinion it is illegible and should be simplified or removed.
Figure 4. Please round p-values ​​to three decimal places. Maybe in the case of a p value lower than 0.05, give letters for homogeneous groups?
Figure 6 could also be corrected, there are no signatures on the horizontal axes. The determination of microorganisms can also be shortened. In general, figures should be self-explanatory.
Author Response
Please see the attachment.

This manuscript is a resubmission of an earlier submission. The following is a list of the peer review reports and author responses from that submission.
Round 1
Reviewer 1 Report
The study on methane emissions in rice fields by the authors is well structured, the discussion responds to the objectives and the bibliography is current. My recommendation is to post it as is.
Reviewer 2 Report
Dear Authors.
In the current version, I cannot support your manuscript for publication. I have a lot of comments. First of all, the emphasis, according to the title of the manuscript, should be on methanogens and methane emissions. The relationship between methanogens and other microorganisms can/should be discussed. Not every representatives of Archaea is a methanogen so if all identified Archaea are discussed it may be worth considering presenting bacteria that support methanogenesis by providing suitable substrates. As we know methanogens occur in consortia with other microorganisms, including bacteria. Furthermore, the manuscript needs to be rebuilt and completed. Language correction is essential. See comments below.
General comments:
Captions tables and figures are too general.
Language correction is necessary.
The title does not match the content. The title suggests that this will be a paper on methanogens, meanwhile all identified Archaea, that are not related to methane formation or this relation is not explained, are extensively discussed and debated.
Authors sometimes confuse metanogens with metanotrophs.
Authors should concentrate on all identified methanogens. In discussion they described more methanogens than in results section.
I found a lot of contradictions and false information about methanogens (e.g. lines 406, 422, 462).
The novelty of the study was not emphasized.
All tables should be place in the text of manuscript.
The Results and Discussion sections need to be restructured to make them related, understandable, and consistent with the results presented. I have indicated some selected examples below.
Particular comments:
Lines 24, 73, 248, 306, 327 and whole manuscript
Use a capital letter for taxonomic unites like Archaea.
Line 26
Methanogens is a common name, write in small letter
Lines 27 and 28, 141, 142,
Use abbreviation after it explanation.
Lines 37-39
Use more recent literature and information. Give also the time frame of comparison.
Line 41-43
Correct the language style.
Line 132, “Analysis of soil physical and chemical properties”
How many replicates for each parameters were made?
Line 145, “Analysis of Soil microorganism’
How many replicates were made?
Line 150
Used primers were dedicated only to amplify the bacterial V3-V4 region?
Line 159
“2.3.4. Determination of CH4 mission flux”. How often the methane flux was measurement? For how many days?
Line 161 and whole manuscript.
Why you used capital letter in word “soil”?
Lines 163-165
“To eliminate the influence of plant photosynthesis and ground litter on the data, the plots were cleaned the day before each measurement without disturbing the surface soil.”
How plants were removed without disturbing the surface?
Line 177
“The changes of heavy metal content in paddy soil after manure application...”
Which methods were used for determination of heavy metals content?
Line 195
“...both PM and OF groups was higher than that of CK group, both were higher by 3.00 g·kg-1, reaching 18.37%” - check with Table 2.
Line 228
“Bacterial richness and diversity of soil archaea in paddy field applied with manure” – were there made a separated analysis for Bacteria and Archaea?
Line 233-235
“The effects of different manure applications on the diversity and abundance of archaeal microorganisms...” - only archaeal?
Line 241-242
“The results indicated that the diversity and abundance of archaea in paddy field were reduced in different degrees with the growth of rice after manure application.” – Were performed separated analysis for Bacteria and Archaea?
Line 251
Why you cited literature? aren't these your results?
Line 254
Start the sentences with capital letter.
Line 323
“Figure 3. Redundant analysis of microbial community gate levels”
This figure consists of 3 graphs, it should be made clear what is shown in each graph.
Line 363
“Figure 4. Univariate anova of intergroup differences in microphylum and genuslevels in paddy 363
fields.” Add precise information what is shown in particular graphs.
Line 286-287
” At the heading stage, the relative abundance of Methanobacterium in the OF group was the highest at 15.0%, which was 5.1% and 7.2% higher than that in the CK and PM groups…”
In which of graph it is presented? Heading stage according to description in line 109 are variants CK3, PM3, OF3. I found no confirmation of this sentence in any of the charts. Correct your description.
Line 353-354
“The relative abundance of Methanobrevibacter in the PM group was significantly higher than that of the OF and PM groups, 36.33 and 21 times higher, respectively.” Please correct.
Line 381
“Figure 5. The effect of manure application on CH4 emission from paddy field soil.” The negative value CH4 emission were measured? Czy mógÅ‚byÅ› wyjaÅ›nić te wyniki w dyskusji? Czy to rezultat jakiegos porównania?
Lines 384-386
” The phyla with relatively high abundance at the phylum level after manure application were Crenarchaeota, Halobacterota, Thermoplasmatota, Altiarchaeota, and Euryarchaeota, and these microorganisms may play an important role in the emission of soil CH4 from rice fields.”
All of this group? Could you explain?
Line 403
“Most methanogens (such as Methanobacterium) use H2 as methanogenic substrate…” In addition to H2, they also need a carbon source in the form of CO2. Consider whether the identified microorganisms include those that can provide specific substrates for the identified methanogens.
Line 403-405
Methanogens belonging to Archaea, so are not bacteria.
Line 406-407
“Methanosarcina, a methanotrophic bacterium, which may be related to the metabolic capacity of methane”
Methanosarcina is not a metanotroph genus and not bacteria.
Line 410
“Wen et al. [20] found that Methanosarcina, Methanosaeta and Methanobrevibacter may contribute to the reduction of CH4 emissions after studying the effect of high temperature compost on methanogenic microbial communities.” Could you explain?
Lines 422-423
“Different environmental factors in paddy soils can affect the CH4 oxidation potential, abundance, and community structure of methane-oxidizing bacteria (Figure 6).
The purpose of your work was to investigate methane oxidation or methane formation?
Line 433, 486, 504
“Methanomassiliicoccus is a methanotrophic methanogenic bacteria.”
Neither a mathanotrophic nor a bacteria.
Lines 462-463
“Rice_Cluster_I is the common methanogens in the paddies rice rhizosphere, which produces H+ during photosynthesis”. Really? Check the correctness of the sentence.
Line 670
“Table 2. Effects of manure application on in paddy soil (g·kg-1)” – use a more informative/proper table caption. Not all parameters are expressing in g kg-1.
Also line 302, caption of figure 2 must be corrected.
Reviewer 3 Report
Land-1567022
Effects of pig manure and its organic fertilizer application on methanogens and methane emission in paddy fields.
Wu et al.
Overall
The use of English would be improved with some careful proofreading. Most errors are small or trivial, but do reduce clarity to some extent. However, most of the manuscript is clear and written well, even when the arguments put forth in the second Discussion section are poorly supported they are fairly clearly articulated.
- Introduction
I would like more description of the key organisms – are all of the Methano- genera known to be methanogens? Are all methanogens Archaea? Were any important bacteria found (such as methanotrophs)?
- Results and discussion
LN241-243 states that diversity and abundance of Archaea decreased with the growth of rice, but in Table 3 all of the values share the same significance letter, a, indicating any differences are not significant. So, the community of Archaea in the paddy fields did not seem to respond to either fertilization or to plant growth, contradicting the statement on lines 241 to 243.
In light of the above comment, I see the word "significant" (or significantly) nowhere in the paragraphs from line 252 to 299. Those paragraphs describe small differences in relative abundance of different groups of Archaea under the different treatments and plant growth stages. Are any of those differences statistically significant?
LN305-306 states "Manure application changed the community composition of soil archaea in rice fields" but cites a separate study – did community composition change in this study? Or only in the cited paper? Furthermore, this sentence continues, stating "was especially significant at the heading and maturity stages." yet no statistical tests are presented, and it is not at all clear what the distinction might be between "significant" and "especially significant".
More care needs to be taken with the word "significant", it has specific meaning when used in a study such as this one and should not be used to emphasise perceived importance or degree of difference.
LN372 shows an example of this – "group decreased more significantly" is not an appropriate use of the word "significantly". I see no p-value or other indication of statistical significance in that sentence.
LN365-379 – were the differences in methane flux significantly different under different treatments and at different times? No indication of statistical tests are given in this paragraph.
- Discussion
If this is the Discussion section, why was the previous section labelled "Results and discussion" and why did the previous section include considerable reference to other studies, as would be found in a Discussion section?
The attempt to link particular genera to patterns of methane flux is unconvincing (LN393-420). Different archaea changed in abundance, yet no information is given about the metabolic activities of these organisms and how a change of abundance of about 10% might lead to a significant increase or decrease in total, net methane flux. I suspect such information is not available for any member of a complex community of organisms, detected by DNA extracted from soil which may include DNA derived from inactive or dead cells.
The subsequent paragraphs (LN422-516), describing correlations between some genera and soil chemical paramaters is fine, though this long list of p-values seems more suited to a Results section.
LN527-528 – "Manure application increased the emission flux of CH4 from paddy soils." – yet no indication of statistical significance among CH4 fluxes has been shown. Was the increase statistically significant? What test was used?
Methanogen relative abundance seems completely unrelated to CH4 fluxes, as demonstrated by the trend towards zero net flux of CH4 in later plant growth stages yet continuing high relative abundance of most of the detected methanogens (LN553-556).
This directly contradicts line 557-559, that states that increasing relative abundance of some methanogens was "also the main reason for the increase in CH4 emission flux".
What relationship, if any, is there between relative abundance of some methanogens and net CH4 flux? Do you have a measure of absolute abundance, or total metabolic activity of these organisms? Flux at the soil surface is the net of both production (e.g. methanogens) and consumption (e.g. methanotrophs) processes and cannot be used in isolation to infer a relationship between only one group of organisms and their community composition.
- Conclusion
As described above, I do not think conclusion (3) is supported by these results. The link between methanogen relative abundance and net CH4 flux has not been demonstrated.